# Talking Points: Describing and Localizing Pixels

**Matan Rusanovsky, Shimon Malnick and Shai Avidan**
Tel Aviv University
{matanru,malnick}@mail.tau.ac.il and avidan@eng.tau.ac.il

## Abstract

Vision-language models have achieved remarkable success in cross-modal understanding. Yet, these models remain limited to object-level or region-level grounding, lacking the capability for pixel-precise keypoint comprehension through natural language. We introduce a novel framework for *pixel level* grounding. The framework consists of two complementary components: a *Point Descriptor* that generates rich, contextual descriptions of individual keypoints, and a *Point Localizer* that regresses precise pixel coordinates from these descriptions. Unlike prior work that relies on templated prompts or keypoint names, our approach produces free-form, coarse-to-fine descriptions that situate keypoints within their visual context. Since there is no available dataset to train such a system, we introduce *LlamaPointInPart*, a carefully curated dataset of 20K+ image-keypoint-description triplets synthesized from multiple vision-language models, capturing multi-scale information from scene-level context to visual features around the keypoint. For cross-category generalization, we optimize the Point Descriptor on AP-10K via GRPO, using the frozen Point Localizer as a reward model to produce descriptions that maximize localization accuracy. To evaluate our results we establish a new evaluation protocol. Instead of comparing the text description produced by our method to the ground truth, we use the localizer to determine how close is the predicted point generated to the ground truth point. Experiments demonstrate superior performance compared to baseline models on LlamaPointInPart. The bidirectional nature of our framework should enable future applications in both keypoint-guided image understanding and language-guided precise localization. Our code and dataset are publicly available at https://matanr.github.io/Talking_Points.

## 1 Introduction

A central challenge in multi-modal learning is bridging the gap between *dense pixel-level visual features* and *semantic natural language*. Although recent models have greatly improved vision–language alignment, they predominantly reason at the *image* or *object* scale, leaving *fine-grained, pixel-level grounding* largely unexplored. As illustrated in Figure 1, this task of precisely describing and localizing individual pixels proves remarkably challenging: while our method doubles the performance of our baseline (OMG-LLaVA) and outperforms the state-of-the-art foundation model ChatGPT-5, human annotations surprisingly perform worse than ChatGPT-5, underscoring the inherent difficulty of this new pixel-level grounding task.

The advent of *Vision–Language Models (VLMs)* such as LLaVA (Liu et al., 2023a) has substantially advanced cross-modal integration by treating visual patches and linguistic tokens uniformly within a transformer. These VLMs excel at tasks like image captioning, visual dialogue, and image-grounded question answering. Recent grounding works have extended these capabilities bidirectionally. Models like SAM (Kirillov et al., 2023), Semantic-SAM (Li et al., 2023a), and Grounding-DINO (Liu et al., 2023b) accept spatial prompts (points, boxes) or language queries to generate segmentation masks and bounding boxes. Conversely, recent VLMs enable both grounded conversation and spatial output generation: DAM (Lian et al., 2025) produces rich descriptions from visual prompts, Groundhog (Zhang et al., 2024c) generates segmentations from textual descriptions, while OMG-LLaVA (Zhang et al., 2024b) unifies both directions, accepting visual prompts (bound-

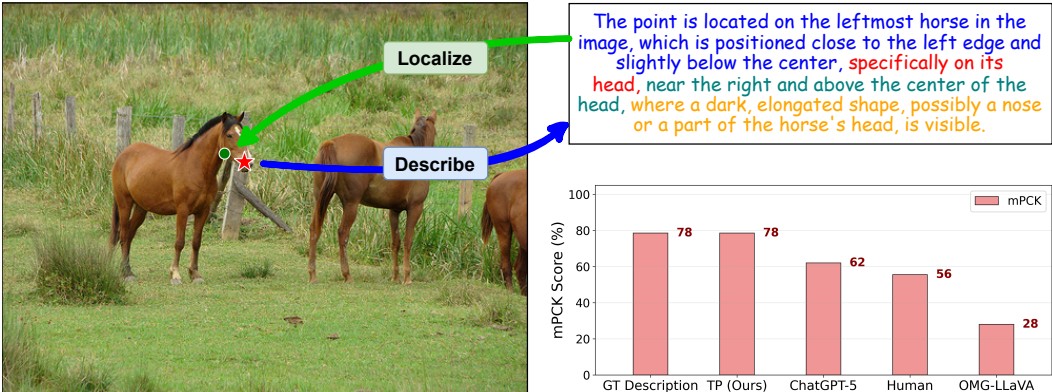

Figure 1: **Talking Points: Describing and Localizing Pixels.** Given an image and keypoint (left, red star), we generate rich descriptions (top right) progressing from scene-level object localization, through part identification, to position within part, and local visual features. We evaluate the descriptions by *localizing* them back to pixels (green point in image). Our TalkingPoints (TP) achieves near GT performance (bottom right), doubling our baseline (OMG-LLaVA) and outperforming ChatGPT-5 and human annotations, which surprisingly perform worst, highlighting the challenge of pixel-level grounding. Evaluation uses *mPCK* (mean Percentage of Correct Keypoints), measuring the fraction of predictions within a normalized distance threshold of ground truth, averaged across fine and coarse thresholds.

ing boxes, masks, points) for region-specific conversations and producing segmentation tokens that decode into spatial outputs through specialized heads. Yet, despite this growing flexibility in bridging vision and language through spatial grounding, all these methods still operate on entire segments or regions rather than reasoning over individual pixels.

Recent efforts such as KptLLM (Yang et al., 2024b) and LocLLM (Wang et al., 2024a) attempt to move beyond object-level prompts toward keypoint comprehension. However, both rely on rigid, template-based textual descriptions tied to predefined anatomy or part labels, falling short of rich, language-grounded localization.

To address these limitations, we introduce two complementary components: a *Point Descriptor* and a *Point Localizer*. The Point Descriptor, given an image and a single pixel (Figure 1, left), generates richly expressive, free-form language that specifies the object's placement within the scene, the part's location within that object, the keypoint's position within that part, and salient visual cues immediately surrounding the keypoint (Figure 1, top right). The Point Localizer then consumes this description to regress the exact pixel coordinate, achieving higher localization accuracy than models relying on templated or name-only prompts. This bidirectional capability, describing pixels in natural language and localizing them back, enables precise pixel-level grounding that significantly outperforms existing approaches.

We first train both components on a carefully curated dataset of 20K+ image-keypoint-description triplets. Our construction pipeline combines part-level annotations with vision-language models operating at different scales, one processing the full image for object-level context and another analyzing masked regions around keypoints for local detail. A large language model synthesizes these complementary perspectives into rich, coherent descriptions that connect precise pixel locations with their semantic context.

We report the results based on a Point Descriptor and a Point Localizer that were trained separately on our curated dataset. Our descriptor-through-localizer evaluation measures description quality through localization accuracy, providing a novel metric for pixel-level language grounding. Additionally, we explore reinforcement learning as a promising direction for extending our approach to novel categories without ground-truth descriptions.

Our contributions are as follows: (1) We construct a dataset of over 20,000 image-keypoint-description triplets with rich natural language capturing multi-scale spatial context; (2) We introduce a **Point Descriptor** and a **Point Localizer** for language-to-pixel mapping; (3) We explore rein-

forcement learning using GRPO as a promising direction for adapting the Point Descriptor to novel categories without ground-truth descriptions; and (4) We propose a novel evaluation methodology measuring descriptor quality through localization accuracy.

## 2 RELATED WORK

### 2.1 KEYPOINT DETECTION AND COMPREHENSION

Traditional keypoint detection methods focus on category-specific models for humans (Lin et al., 2014; Andriluka et al., 2014), animals (Cao et al., 2019; Yu et al., 2021), or objects (Ge et al., 2019), employing either regression-based (Li et al., 2021; Toshev & Szegedy, 2014) or heatmap-based (Xiao et al., 2018; Xu et al., 2022b) approaches. Recent work extends to category-agnostic settings through few-shot keypoint detection (Xu et al., 2022a; Shi et al., 2023) using visual prompts. Several methods, CLAMP (Zhang et al., 2023), X-Pose (Yang et al., 2024a), and CapeX (Rusanovsky et al., 2025), use textual prompts or point explanations for category-agnostic pose estimation. However, these approaches rely on predefined keypoint names and templates rather than free-form descriptions.

The emergence of VLMs has enabled new approaches to keypoint understanding. LocLLM (Wang et al., 2024a) pioneered LLM-based keypoint localization but is trained exclusively on human keypoints, where the model receives textual prompts describing body parts (e.g., "left shoulder," "right ankle") and directly regresses pixel coordinates. While LocLLM incorporates some descriptive context through instruction templates, these remain formulaic and category-specific, limited to human anatomy. KptLLM (Yang et al., 2024b) introduces semantic keypoint comprehension across three tasks: semantic understanding of keypoint names, visual prompt-based detection using support images, and textual prompt-based detection from part names. However, its textual descriptions are generated through a fixed template that combines object category, part name, and keypoint name (e.g., "the left eye of the cat"), lacking free-form, context-rich language that captures the visual appearance or spatial context surrounding the keypoint.

Our work introduces bidirectional keypoint-language grounding: generating rich descriptions from pixel locations and inversely localizing keypoints from these descriptions, enabling true pixel-level language grounding through learned visual context rather than predefined templates.

### 2.2 VISION-LANGUAGE GROUNDING

Vision-language models have evolved from image-level understanding to sophisticated spatial grounding capabilities. Early VLMs like LLaVA (Liu et al., 2023a), BLIP-2 (Li et al., 2023b), and Flamingo (Alayrac et al., 2022) excel at image captioning, visual dialogue, and question answering by treating visual patches and linguistic tokens uniformly within transformers. Recent grounding works have extended these capabilities bidirectionally.

Models like SAM (Kirillov et al., 2023), Semantic-SAM (Li et al., 2023a), and Grounding-DINO (Liu et al., 2023b) accept spatial prompts (points, boxes) or language queries to generate segmentation masks and bounding boxes. Conversely, recent VLMs enable both grounded conversation and spatial output generation: Kosmos-2 (Peng et al., 2023) and Shikra (Chen et al., 2023) innovate by incorporating spatial boxes as inputs and training with region-text pairs for region-level comprehension. DAM (Lian et al., 2025) produces rich descriptions from visual prompts, Groundhog (Zhang et al., 2024c) generates segmentations from textual descriptions, while OMG-LLaVA (Zhang et al., 2024b) unifies both directions, accepting visual prompts (bounding boxes, masks, points) for region-specific conversations and producing segmentation tokens that decode into visual outputs. Ferret (You et al., 2023) and GPT4RoI (Zhang et al., 2024a) further advance region-level visual comprehension through referring and grounding capabilities.

However, these approaches operate at object or segment scales rather than true keypoint-level comprehension. Our work adapts OMG-LLaVA's architecture but fundamentally shifts from object-centric to pixel-centric grounding through Gaussian attention masks, enabling description and localization of individual keypoints rather than entire regions.

## 2.3 REINFORCEMENT LEARNING FOR VISION-LANGUAGE MODELS

Recent advances in reinforcement learning for vision-language models have shifted from subjective human preferences (RLHF) toward spatially-grounded reward signals that provide verifiable, automatic evaluation metrics. Several works establish closed-loop training between description generation and localization. RL-VLM-F (Wang et al., 2024b) uses vision-language foundation models as reward signals based on semantic alignment between descriptions and visual observations. SpatialVLM (Chen et al., 2024) enables dense reward annotation through quantitative spatial understanding, while SE-GUI (Du et al., 2025) implements GRPO (Shao et al., 2024) with self-evolutionary training, computing rewards based on coordinate prediction accuracy.

However, all existing work operates at object or region scales using bounding boxes or segmentation masks as grounding primitives. Our approach uniquely employs pixel-level keypoint localization accuracy as the reward signal, where the Point Descriptor is fine-tuned via GRPO with the Point Localizer serving as reward model. This pixel-centric reward mechanism optimizes for exact coordinate accuracy rather than regional overlap, establishing the first closed-loop training paradigm between keypoint description and localization at the individual pixel level.

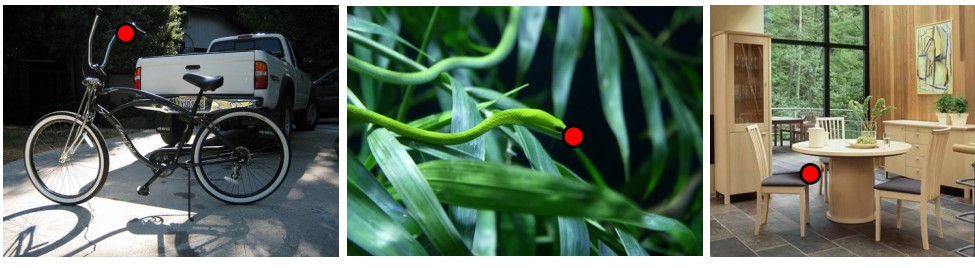

(a) Bicycle handlebar          (b) Snake eye          (c) Chair seat

**(a)** *"The point is on the bicycle, which spans most of the horizontal and vertical axis in the image, specifically on its handlebar, situated to the left and close to the top edge of the bike, and within the handlebar, the keypoint is located slightly above and to the right of the center, in a region that features a curved metal bar with a grip area."*

**(b)** *"The point is located on the snake, which is positioned close to the left edge and slightly above the vertical center of the image, and within this snake, it is situated on the head, specifically near the right edge and below the vertical center of the head, in a region that features a dark, oval-shaped area with a reflective surface, likely the pupil of the snake's eye."*

**(c)** *"The point is located on the chair that is closest to the viewer and positioned on the left side of the dining table, specifically on the chair's seat, which is near the right edge and slightly above the center of the seat, and in the region around the keypoint, there is a small, dark spot standing out against the lighter background."*

Figure 2: LlamaPointInPart dataset examples demonstrating diverse objects and parts across three source datasets: (a) PascalPart116, (b) PartImageNet, (c) ADE20KPart234. Red circles indicate keypoints with corresponding coarse-to-fine descriptions that progress from scene-level object localization, through part identification, to keypoint position within the part, and finally visual features around the keypoint, enabling accurate language-guided keypoint localization.

## 3 METHOD

### 3.1 DATASET CONSTRUCTION

**LlamaPointInPart Dataset** We construct LlamaPointInPart, a high-quality dataset of 20K+ image-keypoint-description triplets, through a multi-stage pipeline leveraging complementary vision-language models (Figure 3). Starting from PascalPart116, ADE20KPart234 (Wei et al., 2023), and PartImageNet (He et al., 2021), we extract images with part-level bounding box annotations. For each image, we compute SIFT (Lowe, 1999) features and select the highest-response keypoint within annotated parts (excluding background). All keypoints in our dataset are semantic by construction: every keypoint corresponds to a meaningful part of an object rather than arbitrary

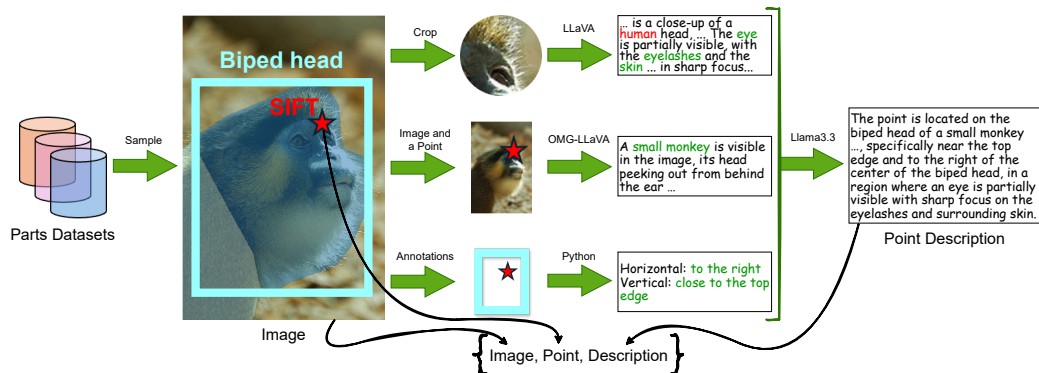

Figure 3: **LlamaPointInPart dataset construction pipeline:** We start with part-annotated datasets, and select the highest-scoring SIFT keypoint within some part (red star). LLaVA generates fine-grained local descriptions from cropped regions around keypoints, while OMG-LLaVA provides object-centric context from full images. Llama3.3 synthesizes these multi-scale perspectives with hierarchical spatial annotations into coherent, coarse-to-fine descriptions, forming our final image-point-description triplets.

background location. Moreover, although SIFT is used to rank candidate points, the final keypoints span both textured and smooth within-part regions, since selection is restricted to annotated semantic parts rather than gradient-rich areas. We determine the keypoint's relative position within its containing part (e.g., "near top edge"), explicitly encoding spatial relationships and instance ordering for disambiguation.

To ensure dataset diversity, we maintain equal proportions across the three source datasets when sampling keypoints, resulting in balanced representation across 64 unique object categories and 297 unique part categories (with some semantic overlap, e.g., "biped" encompassing multiple animal types). Appendix A.1 (Figure 6) visualizes this distribution, with the inner rings showing equal sampling from each source dataset and the outer rings displaying the variety of objects and parts covered. This balanced strategy ensures comprehensive coverage across diverse semantic categories, from animals and vehicles to furniture and household objects.

To capture multi-scale context, we query two VLMs: (1) OMG-LLaVA (Zhang et al., 2024b) receives the image and keypoint to generate object-centric descriptions, and (2) LLaVA (Liu et al., 2023a) processes a Gaussian-masked region centered at the keypoint to extract fine-grained local features. This dual approach captures details beyond part annotations, for instance, identifying a keypoint near a bird's eye despite lacking explicit eye annotations. We synthesize these descriptions via a quantized LLaMA3.3 (Dubey et al., 2024) through a two-stage process (generation followed by refinement) to produce coherent, coarse-to-fine keypoint descriptions. Our descriptions follow a deliberate hierarchical progression: (1) object location within image, (2) part location within object, (3) keypoint position within part, and (4) local visual features. While some descriptions in Figure 3 are abbreviated for space constraints, this coarse-to-fine structure is consistently maintained throughout our dataset.

Our dataset encompasses diverse keypoint types: semantically salient features like the snake's eye pupil (Figure 2b), functional components such as the bicycle handlebar grip (Figure 2a), and seemingly ordinary surface points like the chair seat marking (Figure 2c). This diversity, ranging from visually distinctive landmarks to unremarkable surface locations, ensures our models generalize beyond prototypical keypoints to arbitrary pixel locations, a critical capability for true pixel-level comprehension. We split LlamaPointInPart into 17K training and 4K test examples, maintaining proportional representation across source datasets. We manually tested 5% of the test set samples, and verified that more than 91% of the keypoints can be easily localized using the point descriptions.

**AP-10K Adaptation** To evaluate cross-category generalization capabilities, we leverage AP-10K (Yu et al., 2021), following the experimental split configuration of CLAMP (Zhang et al., 2023) and KptLLM (Yang et al., 2024b). Specifically, we adopt their different order setting, where models are trained on one super-category and tested on another to assess generalization to visually distinct

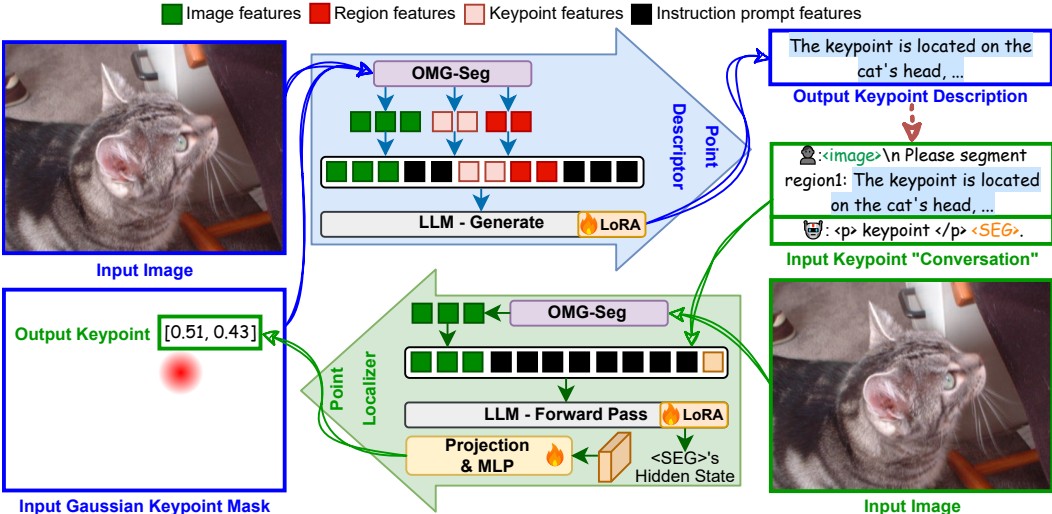

Figure 4: **Talking Point Architecture.** The Point Descriptor (blue) generates textual descriptions from image-keypoint pairs using Gaussian masks centered at keypoints using different type of token features. See legend at top of figure. The Point Localizer (green) regresses keypoint coordinates from image-description pairs through a special <SEG> token that encodes visual information. While the localizer can be used standalone, it can also be applied to the generated descriptions (red dashed arrow) to evaluate localization performance.

animal families. We evaluate bidirectionally: training on Bovidae (22.6K keypoint-image pairs) and testing on Canidae (17K pairs), as well as training on Canidae and testing on Bovidae[1], with both categories annotated with up to 17 keypoints per instance. This setup provides a systematic evaluation of our model's ability to generalize keypoint understanding to unseen taxonomically and visually distinct groups. Since AP-10K provides only keypoint annotations without descriptive context, we utilize these pairs exclusively for reinforcement learning-based fine-tuning (Section 3.4), where our Point Descriptor learns to generate descriptions that maximize localization accuracy.

## 3.2 POINT DESCRIPTOR

Our Point Descriptor adapts OMG-LLaVA's object grounding architecture for pixel-level keypoint description generation (blue part in Figure 4). While OMG-LLaVA predicts segmentation masks to describe entire objects, we replace these with fixed Gaussian attention masks centered at keypoints, fundamentally shifting focus from object-level to pixel-level comprehension.

Given an image $I$ and keypoint coordinates $(x, y)$, we generate a Gaussian mask $M$ centered at the keypoint. The coordinates undergo two parallel transformations through OMG-Seg's decoder: learnable prompt embeddings generate initial semantic queries, while sinusoidal positional encoding of $(x, y)$ followed by linear projection provides spatial information. These initial representations then interact with multi-scale visual features through 9 transformer decoder layers, with the Gaussian mask controlling the attention pattern.

The Gaussian mask constrains the attention mechanism in each decoder layer by defining a boolean attention mask for the cross-attention operation: queries can only attend to image features (keys and values) within the Gaussian region around the keypoint. This differs fundamentally from OMG-LLaVA, where predicted object masks allow attention across entire object boundaries. Our fixed masks force the queries to gather information exclusively from the keypoint's immediate neighborhood.

---

[1]Bovidae includes antelope, argali sheep, bison, buffalo, cow, and sheep. Canidae includes arctic fox, dog, fox, and wolf.

As a result, rather than encoding global object semantics, the queries now capture fine-grained information at the specific keypoint; accordingly, we denote them as *keypoint features*. In OMG-LLaVA, these representations were termed "object features" because they captured object-level information; we rename them here to reflect the shift introduced by our masked-attention modification toward keypoint-level focus. Similarly, the positional encodings, which serve as query positional embeddings throughout the attention operations, become *region features* that anchor spatial reasoning at the keypoint location.

Crucially, while the Gaussian mask constrains attention within the OMG-Seg decoder, the full image features are provided to the LLM alongside the keypoint and region features during description generation (see Figure 4). This ensures the model has access to complete visual context while maintaining focus on the specific keypoint location.

This architectural modification proves essential. Without Gaussian masks (Table 3), performance catastrophically drops: mPCK drops from 78.13 to 23.63, as the model loses the ability to connect specific pixel locations with their descriptions. The refined keypoint and region features are then projected to the language model's embedding space for description generation. We optimize using LoRA adapters (Hu et al., 2022) while freezing the vision encoder, training with standard language modeling loss on LlamaPointInPart descriptions.

### 3.3 POINT LOCALIZER

The Point Localizer inverts the description task: given image $I$ and textual description $D$, it regresses keypoint coordinates (green part in Figure 4). Following OMG-LLaVA's grounding formulation, we structure inputs as: "`<image>\nPlease segment region1: [Description D]`", followed by the response: "`<p> keypoint </p> <SEG>.`", where the special token `<SEG>` encodes visual information. The image is encoded via the vision encoder and projected to the language space through a learned projection using OMG-Seg. These projected features combine with the tokenized prompt and pass through the language model.

We perform a single forward pass and extract the hidden state corresponding to `<SEG>`, $h \in \mathbb{R}^d$. This representation passes through a text-to-vision projection layer, followed by a multi-layer perceptron that maps to normalized coordinates $(\hat{x}, \hat{y}) \in [0, 1]^2$.

Training minimizes the mean squared error between predicted and ground-truth coordinates:

$$\mathcal{L}_{loc} = \mathrm{MSE}(\hat{p}, p_{gt}) \tag{1}$$

where $\hat{p} = (\hat{x}, \hat{y})$ and $p_{gt}$ represent predicted and ground-truth normalized coordinates respectively. We jointly optimize LoRA adapters (Hu et al., 2022) on the language model, the vision-to-text projection layer, and the coordinate regression head. As demonstrated in our ablations (Table 4), the LoRA adaptation of the language model is crucial for effective keypoint understanding.

### 3.4 REINFORCEMENT LEARNING FOR MUTUAL ENHANCEMENT

To enable keypoint comprehension across diverse categories without annotated descriptions, we employ reinforcement learning where the Point Localizer provides reward signals for optimizing the Point Descriptor.

Given an image-keypoint pair $(I, p)$, we sample $G$ descriptions $\{o_1, o_2, \cdots, o_G\}$ from the descriptor policy $\pi_\theta$, where $o_i \sim \pi_\theta(o|I, p)$. For each generated description, the frozen localizer predicts coordinates $\hat{p}_i$. The reward function measures localization accuracy:

$$r_i = -\mathrm{MSE}(\hat{p}_i, p) \tag{2}$$

We optimize the descriptor via modified Group Relative Policy Optimization (GRPO) (Shao et al., 2024). For the sampled descriptions, we compute normalized group-relative advantages:

$$\hat{A}_i = \frac{r_i - \mathrm{mean}(\mathbf{r})}{\mathrm{std}(\mathbf{r})} \tag{3}$$

where $\mathbf{r} = \{r_1, r_2, \ldots, r_G\}$ and apply clipping for numerical stability. Following GRPO, we assign each sequence's advantage to all its tokens and normalize by sequence length. Unlike standard

GRPO, we do not employ importance sampling. The policy gradient objective becomes:

$$\mathcal{L}_{\text{policy}} = -\frac{1}{G} \sum_{i=1}^{G} \hat{A}_i \cdot \frac{1}{|o_i|} \sum_{t=1}^{|o_i|} \log \pi_\theta(o_{i,t}|o_{i,<t}, I, p) \tag{4}$$

where $|o_i|$ denotes the length of description $o_i$ and $o_{i,t}$ represents the $t$-th token in the $i$-th description. This formulation ensures gradient updates are invariant to sequence length.

To prevent policy drift, we incorporate KL regularization against the reference policy $\pi_{\text{ref}}$. Following the GRPO formulation, we compute the KL divergence using an unbiased estimator (Schulman, 2020) at the token level:

$$\mathbb{D}_{\text{KL}}[\pi_\theta \| \pi_{\text{ref}}] = \frac{1}{|o_i|} \sum_{t=1}^{|o_i|} [\exp(r_t) - r_t - 1] \tag{5}$$

where $r_t = \log \pi_{\text{ref}}(o_{i,t}|o_{i,<t}, I, p) - \log \pi_\theta(o_{i,t}|o_{i,<t}, I, p)$ is the log-ratio for token $t$. To prevent gradient instability from extreme probability ratios, we apply conservative clamping to the log-ratio: $r_t = \text{clamp}(r_t, -5, 5)$. This bounds the exponential term to a manageable range while preserving the KL signal. The per-sequence KL divergence is computed by averaging over valid tokens, then averaged across all samples in the batch. The complete training objective combines policy gradient and KL regularization:

$$\mathcal{L} = \mathcal{L}_{\text{policy}} + \beta_{\text{KL}} \cdot \mathbb{D}_{\text{KL}} \tag{6}$$

We implement selective fine-tuning by optimizing LoRA adapters (Hu et al., 2022) while updating only the final two transformer blocks, preserving general linguistic capabilities while adapting high-level representations for keypoint description.

This closed-loop paradigm creates mutual enhancement: the descriptor learns to generate descriptions that maximize localization accuracy, effectively adapting its outputs to the localizer's capabilities. By optimizing descriptions for localizability, we improve the alignment between generated descriptions and the localizer's expected input distribution, enabling strong localization performance on descriptor-generated text.

## 4 EXPERIMENTS

We quantify performance using the Percentage of Correct Keypoints (PCK) metric, where keypoint coordinates are first normalized by image dimensions to $[0, 1]$, then we compute the Euclidean distance between predicted and ground-truth points, counting a prediction as correct if this distance falls below a threshold. We follow (Chen et al., 2025) and use mean PCK (mPCK) that is defined as follows. We average PCK@0.1 and PCK@0.2 to capture both fine-grained accuracy (0.1) and coarse regional localization (0.2) in a unified measure, with full breakdowns in Appendix A.4. This descriptor-through-localizer evaluation extends existing semantic keypoint comprehension tasks by measuring descriptor quality through localization accuracy, emphasizing both expressive language and precise grounding.

### 4.1 SUPERVISED FINE-TUNING

**Point Descriptor and Localizer Training.** We initialize from OMG-LLaVA's pretrained weights and fine-tune the Point Descriptor on LlamaPointInPart's training set for 10 epochs, using a batch size of 8, and a learning rate of $2^{-4}$, optimizing only the language modeling loss $\mathcal{L}_{\text{text}}$ under the same LoRA configuration as OMG-LLaVA (rank 512, effective scaling 0.5, dropout 0.05, no bias). The Point Localizer trains for 15 epochs with learning rate $10^{-5}$ and batch size 8. We optimize LoRA adapters (same as above) on the language model, the vision-to-text projection layer, and the coordinate regression MLP, while freezing all other parameters[2].

---

[2] All training runs in this work were carried out on a single NVIDIA H100 (80 GB) GPU.

Table 1: Performance comparison on LlamaPointInPart test set. Our approach substantially outperforms the OMG-LLaVA baseline, with the Point Descriptor achieving near ground-truth performance.

| Method | mPCK |
|---|---|
| OMG-LLaVA (Zhang et al., 2024b) | 31.03 |
| DAM (Lian et al., 2025) | 42.87 |
| TP (Ours) | 78.13 |
| GT Description | **78.83** |

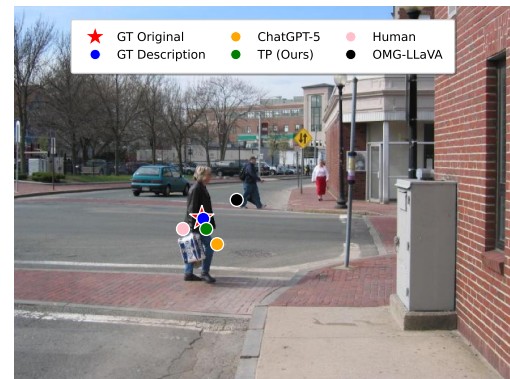

Figure 5: Qualitative keypoints visualization.

**LlamaPointInPart Results.** Table 1 presents localization accuracy on LlamaPointInPart's test set, evaluated using our Point Localizer. Our Point Localizer with ground-truth test descriptions achieves 78.83% mPCK. While with the OMG-LLaVA (Zhang et al., 2024b) and DAM (Lian et al., 2025) baselines, our Point Localizer achieves only 31.03% and 42.87% respectively. Using predicted descriptions from our Point Descriptor maintains robust performance (78.13%), achieving ×2.5 performance boost compared to our OMG-LLaVA baseline. This demonstrates that at the test time of our Point Localizer, we can replace ground-truth descriptions with generated ones while maintaining localization performance.

**Extended Evaluation with Foundation Model and Human Annotations.** We extended evaluation to ChatGPT-5 and human descriptions on 100 test samples. As shown in Figure 1 (bottom right), our approach achieves ground-truth performance (78%), consistently maintaining a performance boost of more than ×2.5 compared to OMG-LLaVA's baseline and surpassing both ChatGPT-5 (62%) and surprisingly, human annotations (56%). Note that these evaluations use a fixed localizer trained on the LlamaPointInPart dataset. The lower human performance may reflect differences in description style from our training data rather than humans' inability to accurately describe keypoint locations. Figure 5 shows a qualitative example. Additional examples of generated descriptions and localizations are presented in Appendix A.2, Table 5.

## 4.2 LOCALIZATION-BASED REINFORCEMENT LEARNING

We evaluate our RL approach for generalizing to novel categories without ground-truth descriptions on the cross-category generalization setup described in Section 3.1[3].

Table 2 reports performance before and after RL adaptation. Although absolute accuracy remains limited due to the challenge of localizing keypoints with rich text on a visually distinct dataset, RL fine-tuning yields consistent improvements. Training on Bovidae and testing on Canidae shows modest gains (∼0.4%), while the reverse setup demonstrates larger improvements (∼6.3%). Overall, these results suggest that localization-based RL represents a promising direction for scaling keypoint understanding by exploiting abundant keypoint-image pairs without costly description annotations.

Table 2: Cross-super-category generalization on AP-10K. RL adaptation shows promising improvements over zero-shot performance.

| Test Set | Method | mPCK |
|---|---|---|
| Canidae | TP (zero-shot) | 29.85 |
| | TP+RL (on Bovidae) | **29.96** |
| Bovidae | TP (zero-shot) | 28.56 |
| | TP+RL (on Canidae) | **30.36** |

## 4.3 ABLATION STUDIES

**Gaussian Mask Guidance.** We investigate the importance of providing explicit Gaussian masks around keypoints during Point Descriptor training. When relying solely on OMG-LLaVA's standard

---

[3]Due to computational constraints, we conducted these RL experiments on a subset of the data. See Appendix A.3 for RL setup and computational considerations.

Table 3: Impact of explicit Gaussian masks. Without visual guidance, the model fails to connect keypoints with descriptions.

| Configuration | mPCK |
|---|---|
| w/o Gaussian mask | 23.63 |
| Point Descriptor | **78.13** |

Table 4: Impact of language model adaptation. Freezing the LLM and training only projection layers severely degrades performance.

| Configuration | mPCK |
|---|---|
| w/o LLM adaptation | 47.60 |
| Point Localizer | **78.83** |

mechanism to predict object masks from input keypoints, without the Gaussian visual guidance, the model completely fails to learn the connection between keypoints and their descriptions (Table 3). This demonstrates that explicit visual marking is crucial for the model to establish spatial-semantic correspondence. This setup also corresponds to fine-tuning the original OMG-LLaVA architecture on LlamaPointInPart, since it retains OMG-LLaVA's object-mask prediction mechanism without our Gaussian modification. Despite being trained on the same data, this adapted OMG-LLaVA variant achieves only 23.63 mPCK (compared to 78.13 mPCK when using the Gaussian mask), indicating that the Gaussian-mask formulation is essential for pixel-level grounding.

**Language Model Adaptation.** Fine-tuning the language model proves essential for keypoint comprehension. Without LoRA adaptation, keeping the LLM frozen while training only projection layers and the regression head, performance drops substantially (Table 4). This confirms that keypoint-specific language understanding requires adaptation of the language model's representations.

## 5 DISCUSSION

**Conclusions.** We presented a framework for pixel-level keypoint comprehension through natural language, introducing a Point Descriptor that generates rich contextual descriptions and a Point Localizer that regresses precise coordinates. Our approach moves beyond templated prompts to produce free-form, coarse-to-fine descriptions that capture multi-scale spatial context. Through reinforcement learning using the frozen Point Localizer as a reward model, we optimize the Point Descriptor to generate descriptions that maximize localization accuracy.

Our method achieves near ground-truth performance on our new LlamaPointInPart and significantly outperforms baseline models, demonstrating the effectiveness of task-specific architectures for pixel-level understanding. The reinforcement learning approach shows promising improvements when generalizing across taxonomically distinct categories in AP-10K. Importantly, this RL approach is particularly promising for scaling, as keypoint-image pairs are substantially easier to collect than the complete image-keypoint-description triplets required for supervised training, opening a path towards training on larger and more diverse datasets.

**Limitations and Future Work.** Our descriptions currently rely heavily on spatial context, requiring the image to remain unchanged, which limits applicability to scenarios like stereo matching or multi-view settings. The analysis revealed that human-annotated descriptions also exhibited similar spatial dependencies, suggesting this may be an inherent characteristic of pixel-level localization tasks where context is crucial to uniquely identify specific points. More challenging still is the task of image correspondence: generating a description from a point in one image that can identify the corresponding point in an entirely different image.

A practical limitation of our current evaluation is that it relies on a single Localizer trained on LlamaPointInPart. As a result, the metric is most reliable for comparing methods that operate within a similar descriptive style. The goal of this work is not to position this Localizer as a universal evaluator, but to offer a consistent, reproducible protocol for the emerging task of pixel-level description–localization. Developing style-robust or human-trained localizers would further expand the generality of the evaluation.

Future work should explore developing multi-view datasets and descriptions that emphasize semantic and appearance-based features over spatial relationships. We hope that releasing our dataset and framework will encourage the community to build upon this direction, ultimately driving progress toward even finer-grained and more reliable localization capabilities in the future.

ACKNOWLEDGMENTS

Part of this research was supported by ISF grant 2132/23.

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

# A    APPENDIX

## A.1    ADDITIONAL DETAILS ON LLAMAPOINTINPART CONSTRUCTION

Figure 6 presents the compositional distribution of objects and parts, and Figure 2 provides representative dataset examples.

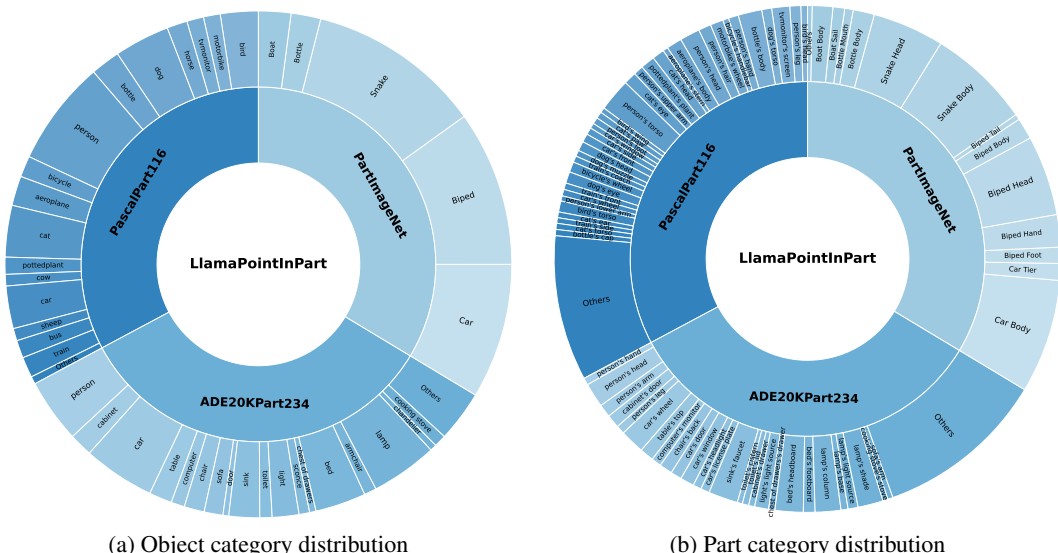

(a) Object category distribution                    (b) Part category distribution

Figure 6: LlamaPointInPart dataset composition showing (a) 64 object categories and (b) 297 part categories across our 20K+ keypoint-description pairs. Inner rings indicate source datasets (Pascal-Part116, ADE20KPart234, PartImageNet), outer rings show sampled objects and parts.

## A.2 Additional Examples

We provide further qualitative examples in Table 5, accompanied by keypoint descriptions from different sources. Each example is shown in a separate column: the top row displays the image with localizations from each source, distinguished by different markers, while the bottom rows present the corresponding textual descriptions from each source.

Table 5: Keypoint Localization Descriptions Comparison.

| | | |
|---|---|---|
| GT Description | The point is located on the cat that spans most of the image, specifically on its paw, which is situated to the right and close to the top edge of the cat, and within this paw, the point is slightly to the right and above the center, in a region featuring a small, dark, circular area with a lighter center, possibly a pupil or a small opening. | The point is located on the rightmost dog in the image, which is close to the right and bottom edges of the picture, and it is positioned on the dog's nose, specifically slightly above and to the right of the center of the nose, with the region around the keypoint being a small, dark, and somewhat circular shape, likely the nose itself. |
| TP (Ours) | The point is located on the cat, which spans most of the horizontal and vertical axis of the image, specifically on the cat's paw, which is near the right and top edge of the cat, and within the paw, the point is slightly to the left and above the center, in a region that features a small, dark, curved shape with a smooth texture. | The point is located on the dog that is positioned closer to the right and bottom edges of the image, and it is the second dog from the left, on its eye, which is situated slightly to the left and above the center of the dog's head, and within the eye, the point is slightly to the left and below the center, in a region that features a small, dark, circular area with a lighter center, possibly a pupil. |
| ChatGPT-5 | The keypoint is located on the raised front paw of the cat, precisely at the outer edge of the paw's upper surface where the fur is dark and contrasts with the bright light shining from the window behind. It sits just above the slightly curved claws that are silhouetted against the background, directly opposite the cat's open mouth and extended tongue. The paw is angled upward, and the keypoint lies on the side facing the camera, slightly to the right of the cat's nose and whiskers, framed by the surrounding purple feather boa that drapes across the cat's chest and lower body. This placement marks a distinct point on the paw's contour, highlighted by the sharp outline against the sunlit backdrop. | The keypoint is located on the smaller brown puppy lying in the snow, specifically at the center of its head where the snout meets the forehead, just above the nose. It sits directly opposite the larger light-colored dog, which is bending its head downward as if sniffing toward the smaller one. The keypoint is positioned slightly right of the image's center, on the puppy whose body is stretched out horizontally on the snow. Surrounding cues include the textured snow surface beneath both dogs, the large blue metal gate in the background, and the clear contrast between the smaller puppy's reddish fur and the white snow. This placement highlights the midpoint of the puppy's face, precisely where its head is directed toward the approaching dog. |
| Human | The point is located on the hand of the cat in the iamge, on the same horizontal line that crosses the middle of the cat's nose, on the middle part of the paw. | the point is located on the tip of the nose of the right (and smaller) dog |
| OMG-LLaVA | The window is clear. | The dog is brown. |

### A.3 REINFORCEMENT LEARNING SETUP AND CONSTRAINTS

**Point Descriptor Reinforced Learning Fine-tuning.** We set the group size $G = 3$, KL-penalty coefficient $\beta_{\text{KL}} = 0.1$, and a learning rate of $5 \times 10^{-6}$. Fine-tuning is conducted for 3 epochs with batch size 10, reusing the same LoRA configuration. This stage explicitly optimizes the descriptor towards producing localization-focused descriptions, complementing the language-only supervised objective.

**Scope and Computational Constraints.** RL training requires generating multiple descriptions per sample in every forward pass, which introduces substantial computational overhead. For this reason, our RL experiments focus on a targeted cross-category setting rather than large-scale domain transfer. While Bovidae and Canidae share similar body plans, this setup isolates whether the descriptor can improve using only (image, keypoint) pairs without ground-truth descriptions. Despite the limited visual divergence, the consistent gains indicate that the pixel-level localization reward is effective for adapting the descriptor beyond its supervised domain. This represents the first application of localization accuracy as a reward for keypoint description, offering a scalable direction for future work on broader category shifts.

## A.4 EVALUATIONS USING PCK@0.1 AND PCK@0.2

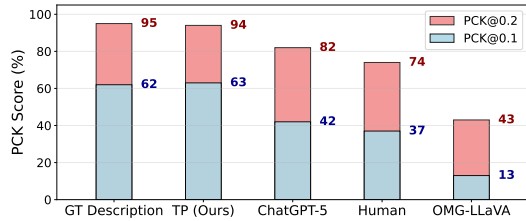

Figure 7: PCK@0.1 and PCK@0.2 breakdown for the mPCK results presented in Fig. 1.

Table 6: PCK@0.1 and PCK@0.2 breakdown for the mPCK results presented in Table 1.

| Method | PCK@0.1 | PCK@0.2 |
|---|---|---|
| OMG-LLaVA | 17.26 | 44.80 |
| DAM | 28.24 | 57.49 |
| TP (Ours) | 63.93 | **92.33** |
| GT Description | **65.60** | 92.05 |

Table 7: PCK@0.1 and PCK@0.2 breakdown for the mPCK results presented in Table 2.

| Test Set | Method | PCK | |
|---|---|---|---|
| | | @0.1 | @0.2 |
| Canidae | TP (zero-shot) | 15.97 | 43.72 |
| | TP+RL (on Bovidae) | **16.16** | **43.76** |
| Bovidae | TP (zero-shot) | 15.49 | 41.62 |
| | TP+RL (on Canidae) | **16.63** | **44.09** |

Table 8: PCK@0.1 and PCK@0.2 breakdown for the mPCK results presented in Table 3.

| Configuration | PCK@0.1 | PCK@0.2 |
|---|---|---|
| w/o Gaussian mask | 11.54 | 35.72 |
| Point Descriptor | **63.93** | **92.33** |

Table 9: PCK@0.1 and PCK@0.2 breakdown for the mPCK results presented in Table 4.

| Configuration | PCK@0.1 | PCK@0.2 |
|---|---|---|
| w/o LLM adaptation | 27.90 | 67.30 |
| Point Localizer | **65.60** | **92.05** |

## A.5 COMPUTATIONAL ANALYSIS

Model Size: The Point Descriptor shares OMG-LLaVA's architecture (7B LLM backbone). The Point Localizer adds only a lightweight MLP head (512-128-2 parameters), representing negligible overhead.

Training Time (single H100 80GB GPU): Point Descriptor fine-tuning: ∼11 hours on LlamaPointIn-Part (17K samples, 10 epochs); Point Localizer: ∼64 hours (15 epochs); RL fine-tuning: ∼22.5 hours on Bovidae subset (22.6K samples), ∼16.8 hours on Canidae subset (17K samples).

Inference Time: Full pipeline achieves approximately 0.35 samples/second on Bovidae (∼18 hours) and 0.39 samples/second on Canidae (∼12 hours).

## A.6 THE USE OF LARGE LANGUAGE MODELS (LLMS)

The text in this paper was refined with the help of LLMs to improve clarity and style. They helped polish the writing.

