# OpenReview forum: "Talking Points: Describing and Localizing Pixels"
_ICLR.cc/2026/Conference — ICLR 2026 Poster_

### Official Review · Reviewer_1BTH · 2025-10-30

**Soundness:** 3
**Presentation:** 3
**Contribution:** 3
**Rating:** 6
**Confidence:** 3

**Summary:**

This paper addresses the core problem of ​​pixel-level vision-language understanding​​ by proposing a novel bidirectional framework, ​​TalkingPoints​​. The framework aims to bridge the gap between dense pixel features and natural language semantics and consists of two complementary components: a ​​Point Descriptor​​, which generates rich, contextual natural language descriptions given an image and a keypoint, and a ​​Point Localizer​​, which regresses precise pixel coordinates from text descriptions. Due to the lack of readily available training data, the authors constructed a large-scale dataset, ​​LlamaPointInPart​​, comprising over 20,000 image-keypoint-description triplets. A key methodological innovation is the use of ​​Gaussian attention masks​​ to achieve pixel-level focus. The paper also explores a ​​Reinforcement Learning (RL)​​ strategy using the localizer as a reward model to optimize the descriptor's generalization to unseen categories. Experiments demonstrate that the method achieves performance close to ground-truth annotations (78.13% mPCK) on the proprietary dataset, significantly outperforming baseline models (e.g., OMG-LLaVA at 31.03%), and shows promise in cross-category generalization tasks. The primary contributions are the proposal of a new task for pixel-level language grounding, a novel method, a new dataset, and a new evaluation protocol.

**Strengths:**

- Novelty of Task and Framework:​​ This is the first systematic exploration of a bidirectional task involving free-form language description and localization of pixel-level keypoints. The proposed TalkingPoints framework is cleverly conceived, with the two components being highly complementary and logically coherent.
- High-Quality Dataset Construction:​​ The pipeline for creating the LlamaPointInPart dataset is rigorous and innovative, synthesizing multi-scale VLMs and a large language model to ensure the richness and diversity of descriptions. The dataset's balance and quality (over 91% localizability) form a solid foundation for reliable experiments.
- Comprehensive Experiments and Analysis:​​ The evaluation design is reasonable. It not only compares performance but also provides in-depth validation of the core components' effectiveness through ablation studies. The inclusion of comparisons against strong baselines and human annotations strengthens the persuasiveness of the results.

**Weaknesses:**

- ​​Generalization and Spatial Dependency:​​ The paper correctly identifies the strong dependence on image spatial context as a major limitation. This severely restricts its applicability in scenarios requiring view or appearance invariance, such as stereo matching or cross-image correspondence.

- ​​Sufficiency of RL Experiments:​​ As mentioned, the RL experiments were conducted on a data subset, making their results preliminary and indicative. Full-scale experiments on larger datasets, with reports of statistical significance, are needed to firmly demonstrate the effectiveness of this approach.

- ​​Computational Cost Analysis:​​ The framework involves two large models (Descriptor and Localizer). The inference speed and computational cost are not discussed, which is an important consideration for practical applications.

**Questions:**

N/A

---

> ### Author Response · Authors · 2025-11-16
>
> We sincerely thank the reviewer for the comprehensive evaluation and recognition of our work's contributions.
>
>
> **Regarding Generalization and Spatial Dependency (Weakness 1)**:
> We appreciate the reviewer's recognition of this important limitation, which we explicitly discuss in Section 5. Our current framework prioritizes precise single-image keypoint localization, which inherently relies on spatial context for disambiguation.  As noted in our future work discussion, extending the framework to view-invariant scenarios (stereo matching, cross-image correspondence) represents an important research direction. This would require developing suitable multi-view datasets, enabling descriptions that emphasize semantic and appearance-based features over spatial relationships.
>
>
> However, interestingly, this reliance on spatial cues was also evident in some of the human-annotated descriptions, suggesting it may be a natural characteristic of pixel-level localization tasks where context is crucial for uniquely identifying specific points. We will expand this discussion in our revision.
>
>
> **Regarding Sufficiency of RL Experiments (Weakness 2)**:
> We recognize the preliminary nature of our RL experiments. As noted in footnote 4 (line 431), we were constrained by computational resources. RL training requires generating multiple descriptions per sample (G=3 in our setup) in each forward pass, introducing substantial computational overhead compared to standard supervised training, making full-scale experiments prohibitively expensive within our resource constraints.
>
>
> Rather than conducting large-scale RL training, we chose a targeted experiment to demonstrate feasibility and potential of the localization-based reward approach for adapting to novel categories without ground-truth descriptions. Our consistent improvements (0.4 and 6.3% mPCK gains) across different category pairs validate the concept. Full-scale RL experiments with comprehensive statistical analysis represent a promising direction for future work, and we will clarify this more explicitly in our revision.
>
>
> **Regarding Computational Cost Analysis (Weakness 3)**:
> We appreciate this practical consideration. Our framework's computational characteristics are as follows:
>
>
> **Model Size**: The Point Descriptor shares the same architecture as OMG-LLaVA (7B LLM backbone). The Point Localizer adds only a lightweight MLP head for coordinate regression (512→128→2 parameters with ReLU activation and Sigmoid output), representing negligible overhead.
>
>
> **Training Time** (single NVIDIA H100 80GB GPU):
> * Point Descriptor fine-tuning: ±11 hours on LlamaPointInPart (17K training samples, 10 epochs)
> * Point Localizer training: ±64 hours (15 epochs)
> * Point Descriptor RL fine-tuning: ±22.5 hours on Bovidae subset (22.6K samples), ±16.8 hours on Canidae subset (17K samples)
>
>
> **Inference Time**: Full pipeline (description generation + localization) achieves approximately 0.35 samples/second on Bovidae (±18 hours) and 0.39 samples/second on Canidae (±12 hours).
>
>
> We will include this analysis in our revision.

---

> ### Author Response · Authors · 2025-11-23
>
> Thank you for your valuable feedback. We have revised the manuscript to address your concerns. Below is a summary of the main changes:
>
> **W1**: We have addressed this concern by elaborating on the similarity to human descriptions. Please see lines 524-530 (specifically 525-529).
>
> **W2**: We have clarified this point with modifications at line 485 and expanded discussion in lines 818-826.
>
> **W3**: We have added discussion addressing this concern. Please see lines 904-914.
>
> Additionally, we have moved **Figure 2** from the appendix to the main text, which presents examples from our LlamaPointInPart dataset.

---

### Official Review · Reviewer_YhME · 2025-10-31

**Soundness:** 4
**Presentation:** 4
**Contribution:** 4
**Rating:** 6
**Confidence:** 3

**Summary:**

This paper introduces a novel framework for bidirectional, pixel-level language grounding, a task that aims to bridge natural language with precise pixel coordinates. The authors propose a two-component system: a Point Descriptor that generates rich, multi-scale descriptions for a given pixel, and a Point Localizer that regresses a pixel's coordinates from such a description. Key contributions include the framework itself, a new synthetic dataset of 20K+ image-keypoint-description triplets named LlamaPointInPart, and a reinforcement learning approach for generalizing the descriptor to new categories without textual annotations.

**Strengths:**

- The paper introduces a novel problem formulation by shifting vision-language grounding from the common object/region level to the more precise pixel level. This addresses a clear gap in the literature. The creation and open-sourcing of the LlamaPointInPart dataset is an appreciated contribution to enable research on this new task.
- The proposed two-component framework is technically sound and logically structured for the bidirectional task. The adaptation of an existing architecture via a fixed Gaussian attention mask is a straightforward method to focus the model on pixel-level features. The empirical results show that the model performs well on the proposed dataset.
- The paper is structured clearly, with a well-defined problem statement, methodology, and experimental setup. The paper is easy to follow, and the provided figures help to illustrate the proposed architecture and data generation process.

**Weaknesses:**

1. **Critical Evaluation Bias:** The primary weakness lies in the evaluation methodology. The "Descriptor-Through-Localizer" protocol is inherently biased, as it measures a description's quality based on its compatibility with a single Point Localizer model. This localizer was trained exclusively on the synthetic LlamaPointInPart dataset, learning its specific "machine dialect." This bias is further compounded by the RL fine-tuning, which explicitly optimizes the proposed descriptor to align with this specific localizer. Consequently, the claims of superiority over other models are not justified. This is also illustrated by the surprisingly low performance of human and ChatGPT-5 descriptions (Fig 1 & 7), which likely reflects a stylistic mismatch with the localizer rather than an inherent inferiority of the descriptions themselves.
2. **Limited Generalization in RL Experiments:** The experiments on cross-category generalization, while a promising direction, have significant limitations.
    - **Scope:** The generalization task was performed between Bovidae and Canidae—two categories of four-legged mammals with similar body plans. This does not provide strong evidence that the RL method can generalize to truly novel and visually dissimilar categories (e.g., from an animal to a bicycle).
    - **Performance:** The reported mPCK improvements from RL are modest, therefore questioning the robustness and scalability of the approach for more challenging generalization scenarios.
3. (minor)**Architectural Constraint of Fixed Attention:** A potential architectural limitation is the reliance on a fixed, symmetric Gaussian attention mask in the Point Descriptor. This assumes the most relevant context is always in an isotropic region around a keypoint. This assumption may not hold for points on elongated structures, thin objects, or part boundaries, where a more flexible and content-aware attention mechanism could be more effective.

**Questions:**

1. Regarding the evaluation bias: Have you considered fine-tuning separate “Point Localizer” models on human-generated or GPT5-generated descriptions (or any other evaluation protocol)? This would create a fairer comparison by evaluating each description source with a model adapted to its specific linguistic style and could provide a more accurate measure of their relative effectiveness.
2. Regarding the RL approach: Could you comment on the scalability of the RL method for generalization? How do you expect it would perform when fine-tuning on a category like "animals" and testing on a visually disparate category like "vehicles" or "furniture"?
3. The dataset construction relies on SIFT keypoints, which are inherently biased towards high-texture regions. How does the model perform when tasked with describing arbitrary points on smooth or textureless surfaces, and how might this dataset bias affect its real-world applicability?

---

> ### Author Response · Authors · 2025-11-16
>
> We thank the reviewer for the thorough evaluation and constructive feedback on both the strengths and limitations of our work.
>
>
> **Regarding Critical Evaluation Bias (Weakness 1, Question 1)**:
> We appreciate this important observation about evaluation methodology. Our goal in this work is to propose a bidirectional framework for pixel-level description and localization, rather than to introduce a definitive benchmark for keypoint descriptions. Accordingly, we do not claim that our Point Localizer is an oracle evaluator; the performance of the overall system is inherently tied to the quality and training distribution of this localizer. Following the reviewer's first question, a natural direction for future work is to improve the localizer, either architecturally or via more diverse and human-centered data. We will discuss this issue in the paper.
>
>
> We agree that the fairest way to evaluate human or ChatGPT-style descriptions would be through localizers explicitly trained or adapted to those description styles. However, collecting sufficiently large-scale human or high-quality LLM annotations for each style is currently labor-intensive and expensive. In this sense, a truly "oracle" localizer that is agnostic to stylistic variability remains an aspirational objective, and we view our work as establishing a quantitative evaluation protocol / baseline for this new task.
>
>
> Given the difficulty of building separate datasets and dedicated localizers for every descriptive model or style, we fixed a single localizer trained on LlamaPointInPart to enable consistent evaluation. We attempted to diversify the training data as much as possible (Section 3.1) by combining multi-scale VLM perspectives, though coverage is naturally constrained by scale and synthesis choices. Our strong results (78.13% mPCK, closely matching ground-truth descriptions at 78.83%) demonstrate that this approach successfully enables pixel-level grounding across diverse objects and parts. We anticipate that subsequent work on this emerging task will build upon our framework by either (i) developing stronger, more style-robust localizers, (ii) designing descriptors that generalize across description styles, or (iii) expanding and refining the underlying data to create more comprehensive evaluation protocols.
>
>
> **Regarding Limited Generalization in RL (Weakness 2, Question 2)**:
> The reviewer raises valid concerns about the scope and magnitude of RL improvements. We offer several perspectives that will be added to the paper.
> 1. **Task difficulty**: Our RL setup is intentionally challenging: we fine-tune on one subset of the dataset (without ground-truth descriptions) and evaluate on a different subset with distinct visual characteristics. The **consistent** improvements (0.4% and 6.3%) validate that the approach successfully adapts to novel categories.
> 2. **Key contribution**: This demonstrates that the descriptor can improve **without expensive (image, keypoint, description) triplets**: using only abundant (image, keypoint) pairs. This is valuable for scaling to new domains where description annotation is impractical.
> 3. **Current constraints**: We used conservative hyperparameters that allow limited change (small learning rate, frozen localizer, limited training) to maintain stability. Longer training and more sophisticated optimization could yield stronger improvements.
> 4. **Future directions**: For the challenging cross-domain scenario (animals to vehicles/furniture) suggested by Question 2, we envision an **iterative co-training approach**: (i) RL improves the descriptor, (ii) improved descriptions generate better pseudo-labels, (iii) localizer fine-tunes on these signals, (iv) stronger localizer serves as improved reward model for RL. This could progressively bridge domain gaps. Given computational constraints and since this wasn't our primary focus, we reserve this for future work.

---

> ### Author Response · Authors · 2025-11-16
>
> **Regarding Architectural Constraint of Fixed Attention (Weakness 3)**:
> We appreciate the reviewer's insightful observation about the potential limitations of fixed, isotropic attention. We clarify that our architecture incorporates substantial flexibility for content-aware attention through its design.
>
>
> While we use a Gaussian mask as a spatial prior, this mask only affects the generation of keypoint features within the OMG-Seg decoder. Crucially, as shown in Figure 3's legend, **image features representing the full visual context are provided to the LLM alongside keypoint and region features during description generation**, though we acknowledge this architectural detail is not sufficiently emphasized in the main text. This design enables the model to learn adaptive attention patterns based on both the localized keypoint information and the complete surrounding visual content, achieving learned, content-aware attention rather than purely geometric masking.
>
>
> The Gaussian mask provides a soft inductive bias that stabilizes training while the model learns these adaptive patterns. Our ablation (Table 3) demonstrates its value: performance drops dramatically from 78.13% to 23.63% mPCK without this guidance. Nevertheless, the reviewer's point about spatially content-aware mechanisms represents an interesting direction for future work. Currently, our keypoint features encode the keypoint and its immediate isotropic neighborhood, and the LLM then contextualizes this information relative to the full image features during generation. A potentially more effective approach would be to generate content-aware keypoint features directly, allowing the feature extraction itself to adapt spatially based on local structure (e.g., elongating along thin objects or aligning with part boundaries). This could provide richer initial representations for challenging cases where the optimal attention pattern deviates strongly from the Gaussian prior.
>
>
> We will clarify the complete information flow, including how image features are provided to the LLM, more explicitly in our revision.
>
>
> **Regarding SIFT Bias (Question 3)**:
> The reviewer correctly identifies that SIFT tends to favor textured locations. However, in our setup the selection is constrained to annotated parts, which are not necessarily high-texture regions. In practice, we observe that many of the resulting supervision points lie on relatively smooth, low-texture surfaces (as discussed in Section 3.1, lines 249-254). For example, the chair seat in the right panel of Figure 6(c) and the cat's paw in the left column of Table 5 both feature ground-truth keypoints located in low-texture regions. This suggests that, although SIFT is used to rank candidates, the final distribution of keypoints spans a variety of within-part locations, not just strong gradients or corners.
>
>
> Because SIFT is only used offline to select supervision points (and is never used as an input feature at training or test time), we did not find it necessary to explicitly introduce additional variance by alternating the keypoint selection method. We expect that the natural variability of within-part locations: ranging from textured edges to smooth interiors, already encourages generalization to real-world scenarios where users may select both textured and low-texture regions. We will elaborate on this point in the text.

---

> ### Author Response · Authors · 2025-11-23
>
> Thank you for your thoughtful review. We have carefully revised the manuscript to address your concerns. Below is a summary of the main changes:
>
> **W1&Q1**: We have addressed this concern with additional analysis in lines 531-536. These additions come together with our existing discussion in lines 459-463, where we note that "these evaluations use a fixed localizer trained on the LlamaPointInPart dataset. The lower human performance may reflect differences in description style from our training data rather than humans’ inability to accurately describe keypoint locations."
>
> **W2&Q2**: We have clarified this point with modifications at line 485 and expanded discussion in lines 818-826.
>
> **W3**: We have added clarification regarding this concern. Please see lines 331-335.
>
> **Q3**: We have addressed this question with additional details in lines 214-238 (specifically 236-238).
>
> Additionally, we have moved **Figure 2** from the appendix to the main text, which presents examples from our LlamaPointInPart dataset.

---

### Official Review · Reviewer_psBk · 2025-11-01

**Soundness:** 3
**Presentation:** 4
**Contribution:** 3
**Rating:** 6
**Confidence:** 4

**Summary:**

This paper proposes Talking Points (TP), a bidirectional framework for pixel-level grounding that (i) generates free-form, coarse-to-fine keypoint descriptions from an image + keypoint (Point Descriptor) and (ii) localizes a keypoint from an image + description (Point Localizer). It introduces LlamaPointInPart, a 20k+ triplet dataset (image, keypoint, description) synthesized from multiple VLMs and fused by an LLM, and evaluates descriptions via localization accuracy using mPCK. The system attains ~78 mPCK, substantially exceeding OMG-LLaVA/DAM baselines on the new test set, and explores GRPO-style RL that optimizes the descriptor using the localizer as a reward model. The authors report modest cross-category gains on AP-10K with RL and discuss limitations around description style bias and spatial dependency.

**Strengths:**

1. LlamaPointInPart composes multi-scale descriptions by querying OMG-LLaVA for object context and LLaVA for local appearance, then LLM-synthesizing coherent coarse-to-fine text; balanced sampling spans 64 objects/297 parts.
2. Unlike prior region/part-level grounding, TP tightly couples language-point mapping with a description-through-localizer metric and architecture tailored to individual pixels.

**Weaknesses:**

1. OMG-LLaVA is capable of segmenting object masks, since its training data is for segmentation. The LlamaPointInPart dataset and TP architecture are tailored for point localization. The direct comparisons between OMG-LLaVA and TP are unfair. Comparing TP with a fine-tuned OMG-LLaVA on LlamaPointInPart is valuable.
2. For keypoint selection, the authors use SIFT with the highest response for keypoint generation. The keypoint validation is necessary but not explained. SIFT detector may choose a point without concrete semantics.  The ratio of semantically rich keypoint in the dataset could be provided.
3. The Python-generated description of the point location is based on the relative position to the part segmentation. This may be confused with the relative position to the edge of an image.

**Questions:**

1. Does TP have the ability to conduct semantic segmentation? How is the performance?
2. Why does the point localizer only use one kind of features of the OMG-Seg?

---

> ### Author Response · Authors · 2025-11-16
>
> We thank the reviewer for the thoughtful feedback and insightful observations.
>
> **Regarding fairness of OMG-LLaVA comparison (Weakness 1)**:
>
> We appreciate this important clarification opportunity. The reviewer points out that comparing TP to OMG-LLaVA may be unfair since OMG-LLaVA was not fine-tuned on LlamaPointInPart. We address this concern by noting two possible fine-tuning scenarios: (1) segmentation mechanism and (2) description mechanism.
> Since we shifted the objective from segmentation to localization, comparing OMG-LLaVA's segmentation to TP's localization is irrelevant (and thus not included). In all our experiments, we use a **fixed Point Localizer** trained on our dataset to evaluate descriptions from OMG-LLaVA (frozen in Table 1), TP, and other methods. This ensures that all description approaches are compared under exactly the same localization mechanism.
>
> Regarding fine-tuning the description mechanism of OMG-LLaVA on LlamaPointInPart: **this setup is already captured by the ablation in Table 3**. The "w/o Gaussian mask" configuration predicts object masks as OMG-LLaVA does while retaining the original OMG-LLaVA architecture. This implements exactly the setup requested by the reviewer (OMG-LLaVA fine-tuned on LlamaPointInPart). The dramatic performance drop (78.13% to 23.63% mPCK) demonstrates that our Gaussian mask modification is crucial for pixel-level grounding, even when trained on the same data. We will clarify the text with respect to this issue.
>
> **Regarding Q1 - Semantic segmentation capability**:
> Instead of producing a segmentation mask given an image and a text description, our Localizer outputs a pair of keypoint coordinates. Our goal is to shift the objective from dense segmentation to point-level description and localization: a different, more fine-grained task. For semantic segmentation, existing specialized models such as OMG-LLaVA/OMG-Seg remain more appropriate.
>
> **Regarding Q2 - Why Localizer uses only one feature type**:
> The keypoint and region features are derived from the input keypoint mask, which is only available to the Descriptor since it receives the selected point as input. By design, this information is not accessible to the Localizer, whose role is to recover the keypoint from image and text alone.
>
> The Localizer therefore uses OMG-Seg only as an image encoder and does not consume keypoint/region features. This separation is intentional: the Descriptor can rely on explicit keypoint/region cues, while the Localizer must infer location purely from image and description, which is exactly the task we aim to evaluate.
>
> **Regarding semantic richness of SIFT keypoints (Weakness 2)**:
> We thank the reviewer for raising this concern. Importantly, **all keypoints in our dataset are semantic by construction** and we will clarify it in the text. As stated in Section 3.1 (lines 206-207): "we compute SIFT features and select the highest-response keypoint **within annotated parts (excluding background).**" This ensures every keypoint corresponds to a meaningful part of an object.
> Figure 6 demonstrates diverse examples from visually distinctive landmarks (snake's eye pupil) to functional components (bicycle handlebar) to ordinary surface points (chair seat). This diversity ensures generalization beyond prototypical keypoints.
>
>
> **Regarding Python-generated spatial descriptions (Weakness 3)**:
> We appreciate this clarification opportunity. Our descriptions include **both image-level and part-level spatial context in a hierarchical progression**: (1) object location within image, (2) part location within object, (3) keypoint position within part, and (4) local visual features.
>
> Figure 6 demonstrates this multi-scale structure. For example, in 6(a): "The point is on the bicycle, which spans most of the horizontal and vertical axis **in the image**" (image-level), then "specifically on its handlebar, situated to the left and close to the top edge **of the bike**" (object-level), followed by part-level and local details. We abbreviated some descriptions in Figure 2 for space constraints, which may have obscured this hierarchical structure. We will clarify this in the revision.

---

> > ### Author Response · Authors · 2025-11-23
> >
> > Thank you for your constructive feedback. We have revised the manuscript to address your concerns. Below is a summary of the main changes:
> >
> > **W1**: We have addressed this concern in the revised manuscript. Please see lines 498-502.
> >
> > **W2**: We have expanded this section with additional details and clarifications. Please see lines 214-238 (specifically 214-236).
> >
> > **W3**: We have clarified this point in the revision (lines 253-258). Additionally, we have moved **Figure 2** from the appendix to the main text, which presents examples from our LlamaPointInPart dataset and exemplifies the descriptions and structure of our dataset samples.
> >
> > **Q1&Q2**: We have addressed these questions in our response above and believe the current manuscript text is sufficient.

---

### Author Response · Authors · 2025-12-02
**General Comment for the New Area Chair**

Dear Area Chair,

Thank you for taking on this paper following the OpenReview incident. To facilitate your review, here's a structured summary of the review process:

**Paper Overview**: TalkingPoints introduces a bidirectional framework for pixel-level vision-language grounding with two components: a Point Descriptor that generates rich natural language descriptions from image-keypoint pairs, and a Point Localizer that regresses precise pixel coordinates from descriptions. We contribute the LlamaPointInPart dataset (20K+ triplets) and demonstrate 78.13% mPCK performance.

**Review Status & Reception**:
* Pre-discussion scores were unanimously positive: 6, 6, 6
* Reviewers acknowledged the novelty of addressing pixel-level grounding (vs. traditional object/region-level), the high-quality dataset construction enabling research on this new task, and the technically sound bidirectional framework design

**Manuscript Revisions**: We have addressed all reviewer concerns with targeted revisions throughout the manuscript. All changes are marked with underlined red text for easy identification. Additionally, we moved Figure 2 from the appendix to the main text for clearer dataset illustration (this is the only unmarked change).

**Main Review Themes & Our Responses**:
1. **Evaluation Methodology & Baseline Comparisons** (YhME, psBk)
    * Reviewer YhME  asked about "Localizer evaluation bias": We clarified that lower human/ChatGPT performance reflects style differences rather than description quality (lines 459-463) and positioned our protocol as a baseline for emerging task rather than universal evaluator (lines 531-536).

    * Reviewer psBk questioned whether our baseline comparison was fair. We clarified that Table 3's "w/o Gaussian mask" configuration is exactly OMG-LLaVA fine-tuned on LlamaPointInPart (as the reviewer requested), and the dramatic performance gap (23.63% vs 78.13% mPCK) validates our architectural contribution (lines 498-502).

2. **RL Generalization Scope** (YhME, 1BTH)
    * Reviewers expressed concerns about the limited scope of RL experiments. We acknowledged computational constraints preventing full-scale experiments and clarified that RL was an exploratory component rather than our primary focus. Despite these limitations, we demonstrated proof-of-concept for adaptation without ground-truth descriptions, achieving consistent 0.4-6.3% improvements (lines 485, 818-826).

3. **Technical Issues or Design Choices** (All reviewers)
    * Reviewers raised concerns about SIFT potentially selecting non-semantic points (psBk) and bias toward high-texture regions (YhME). We clarified that all keypoints are semantic by construction since SIFT selection is constrained within annotated parts only, ensuring 100% semantic validity, and demonstrated that our dataset includes diverse texture levels from smooth surfaces to textured regions (lines 214-238).
    * Reviewer YhME noted potential limitations of fixed attention. We clarified that full image features are provided to the LLM alongside keypoint/region features, ensuring complete visual context despite the Gaussian mask constraint for keypoint features (lines 331-335).
    * Reviewer 1BTH identified strong spatial dependency as a limitation (as we mentioned in the original version). We acknowledged this as characteristic of single-image localization tasks and noted that human descriptions show similar patterns (lines 524-530).
    * Reviewer 1BTH requested computational analysis. We added comprehensive analysis of model size, training time, and inference speed (lines 904-914).
    * Reviewer psBk noted potential confusion between relative positions to part segmentation vs. image edges. We clarified that our descriptions include hierarchical multi-scale context: image-level → object-level → part-level → local features, avoiding ambiguity (lines 253-258), and moved Figure 2 to the main text to better exemplify this structure.

---

### Meta-Review · Area_Chair_EcnM · 2025-12-28

**Summary:**

Reviewers confirmed the novelty, thorough experiments and writing of this paper. After rebuttal and discussion, most of the concerns raised by reviewers are resolved the authors and all the ratings are positive. I thereby recommend to accept this submission.

**Reviewer Concerns:**

The major concerns about evaluation methodology, performance comparisons, generalization scope and design choices were addressed by authors during rebuttal.

**Reviewer Scores:**

All major concerns raised by reviewers were responded during the discussion phase, and I think that the reviewers would like to maintain positive scores.

---

### Decision · Program_Chairs · 2026-01-26

Accept (Poster)